# Mental Health Stigma and Help-Seeking Behaviors Among Primary Healthcare Physicians in Oman

**DOI:** 10.3390/ijerph22070983

**Published:** 2025-06-23

**Authors:** Tharaya Al-Hashemi, Tamadhir Al-Mahrouqi, Salim Al-Huseini, Muna Al Salmi, Rahma Al Nuumani, Fatma Al Balushi, Al Khatib Al Saadi, Muna AlKalbani, Sachin Jose, Samir Al-Adawi

**Affiliations:** 1Al Masarra Hospital, Ministry of Health, Muscat 123, Oman; 2Department of Behavioral Medicine, Sultan Qaboos University Hospital, Muscat 123, Oman; 3Department of Psychiatry, Sohar Hospital, Ministry of Health, Muscat 123, Oman; 4Computing Department, Muscat College, Muscat 123, Oman; 5Research Department, Oman Medical Specialty Board, Muscat 123, Oman; 6Department of Behavioural Medicine, College of Medicine and Health Sciences, Sultan Qaboos University, Muscat 123, Oman

**Keywords:** mental disorders, stigma, help-seeking behavior, physicians, attitude to health, Oman

## Abstract

Background: In Oman, primary healthcare physicians (PHPs) are often the first point of contact in the healthcare system. Understanding the prevalence and impact of stigma among these professionals is crucial to fostering a supportive work environment and promoting access to mental health care. This study evaluated mental health stigma and its association with help-seeking behaviors among PHPs in Muscat, Oman. Methods: A cross-sectional analytical study was conducted from March to May 2023 using cluster random sampling to recruit 191 PHPs. Participants completed a structured questionnaire that evaluated demographic and clinical characteristics, help-seeking behaviors, and perceived stigma. The PPSS developed for this study underwent expert review, pilot tests, and reliability analysis. Data were analyzed using descriptive statistics, Chi-square tests, and multivariate logistic regression, with a significance set at *p* < 0.05. Results: Most of the participants were women (78.5%), aged 30–39 years (49.7%), and Omani nationals (71.2%). More than half (57.6%) reported experiencing depressive episodes, yet only 21.8% sought professional help. High levels of stigma were associated with reluctance to seek professional mental health support, and 24.6% of participants preferred not to seek help at all. Those in the stigma group were significantly more likely to rely on family or friends for support (adjusted OR = 2.873; 95% CI = 1.345–6.138; *p* = 0.006). Common barriers to help-seeking included a lack of belief in the effectiveness of treatment (23.0%) and concerns about confidentiality (19.9%). Conclusions: Mental health stigma remains a widespread problem among primary healthcare physicians in Oman, influencing their behavior and preferences. Interventions to reduce stigma and address barriers to mental health care, such as enhanced confidentiality safeguards and treatment skepticism, are critical to improving physician well-being and healthcare delivery. This study can inform policy and training programs aimed at improving physician well-being and patient care.

## 1. Introduction

Mental health stigma refers to negative attitudes, beliefs, and behaviors directed at individuals with mental health problems. It is often conceptualized through frameworks that identify various components, such as labeling, stereotyping, separation, status loss, and discrimination [1]. Link et al. [2] propose that stigma arises when a person is labeled in a way that associates them with undesirable characteristics, leading to societal prejudice and discrimination against them. This stigma can manifest in different forms, affecting not only those with mental illnesses but also their families and communities. Various authors have identified various aspects, including those arising from the negative perceptions held by society regarding people with mental illness. This public stigma could encapsulate the idea that individuals with mental disorders are dangerous or incompetent [3]. In addition to public stigma, there is self-stigma, which occurs when individuals internalize public stigma, leading to feelings of shame and diminished self-worth. Those affected may believe that they are to blame for their condition or feel unworthy of help [4]. The third type of stigma is known as structural stigma, which entails systemic barriers created by policies and practices that limit opportunities for individuals with mental illness. Examples include inadequate funding for mental health services and discriminatory workplace practices [5]. The final type of stigma is known as perceived stigma, which refers to an individual’s perception of societal attitudes toward mental illness and can influence their behavior and willingness to seek help [1].

Mental health stigma is prevalent in all strata of society, including among healthcare professionals, which, in turn, has the potential to impact both their interactions with patients and their mental health. Studies indicate that healthcare providers often have negative attitudes toward people with mental illnesses, which can lead to discriminatory practices in clinical settings [6]. Several studies have alluded to the view that while healthcare professionals may be more exposed to mental health issues in education and practice, this does not necessarily translate into reduced stigma [7]. Rather, many still exhibit significant biases toward patients with mental disorders. Furthermore, research shows that healthcare workers can experience stigma related to their mental health struggles, leading to a reluctance to seek help [8]. The culture within healthcare settings often discourages openness to mental health struggles due to fears of judgment and professional repercussions. This phenomenon is compounded by “therapeutic pessimism,” where providers believe that recovery for patients with mental illnesses is unlikely, which can also affect their self-perception and willingness to seek help [8]. Some healthcare professionals tend to seek help from informal support systems, including family and friends, and complementary and alternative healing systems [9].

It has been indicated that there are insidious negative repercussions of mental health stigma in healthcare settings among healthcare professionals. Healthcare professionals may fear that seeking mental health support could jeopardize their careers. Concerns about being perceived as incapable or unfit for their roles can lead to the complete avoidance of help-seeking altogether [10]. This has enormous implications for the integrity of healthcare services, including practitioners who have unwarranted burnout, which in turn reduces productivity and most significantly compromises patient care [11].

Oman’s healthcare system has made significant strides in recent decades, characterized by a robust public health infrastructure that aims to meet the needs of its growing population. Primary healthcare physicians play a crucial role in this system, serving as the first point of contact for patients and providing comprehensive medical services, including mental health care. However, mental health facilities are often limited, particularly in rural areas, creating accessibility challenges for those in need of care. The integration of mental health services into primary care remains inadequate, which can exacerbate stigma and hinder effective treatment of mental health conditions. Previous studies have suggested that there is a high frequency of burnout and poor mental health outcomes among doctors ‘tomorrow’. Studies indicate that mental health conditions are not uncommon among healthcare professionals in Oman. However, limited research on mental health stigma and its effects on healthcare professionals in Oman has been carried out.

Numerous tools have been developed to measure stigma related to mental health, yet many of these instruments exhibit significant limitations. Fox et al. [1]. have reported a systematic review of the literature and identified more than 400 stigma measures; however, a substantial portion of these lack rigorous psychometric validation, particularly in terms of cross-cultural applicability and the ability to capture nuanced experiences of stigma. They reported common issues, including the inadequate measurement of actual behavior, cultural relevance, and complexity of stigma constructs. There is a domain-specific need for a culturally relevant scale for perceived stigma among physicians in Oman.

Oman’s unique cultural context requires the development of a culturally relevant scale to measure perceived stigma among healthcare professionals. Current tools may not adequately address the specific attitudes and beliefs prevalent in Omani society regarding mental health. A customized instrument could help capture context specifics, enhancing understanding of the barriers Omani physicians face when addressing their own or their patients’ mental health issues.

The study aimed to evaluate mental health stigma and its impact on help-seeking behaviors among primary healthcare physicians in Muscat, Oman. It sought to measure the prevalence and dimensions of perceived stigma, including devaluation, discrimination, stereotype endorsement, and secrecy while exploring the sociodemographic and clinical characteristics associated with different levels of stigma. Furthermore, the study examined the relationship between stigma and help-seeking behaviors, identifying preferred sources of support, barriers to seeking care, and factors influencing physicians’ willingness to seek professional mental health assistance. To achieve these objectives, the study developed a context-specific stigma assessment tool tailored to capture the unique nuances of mental health stigma within the target population. Our tool was developed to capture physician-specific domains, such as concerns about career jeopardy, professional reputation, and therapeutic pessimism, which are inadequately addressed in existing instruments.

## 2. Materials and Methods

### 2.1. Study Design and Setting

This was a cross-sectional analytical design to assess mental health, stigma, and help-seeking behaviors among primary healthcare physicians. The research was carried out between March and May 2023. A cluster random sampling method was used to ensure a representative selection of participants from various primary healthcare settings in Muscat, Oman.

The Oman health system is structured into three levels: primary, secondary, and tertiary care. Primary care is provided through local health centers, which provide preventive and basic curative services as the first point of contact for most residents. Secondary care is available in the district and regional hospitals, offering more specialized services for cases referred from primary care. Tertiary care is provided by major hospitals in urban areas, providing advanced treatments for complex cases requiring specialized and sub-specialized expertise. This tiered system ensures a comprehensive approach to health needs in the population.

In the Omani healthcare system, primary healthcare physicians (PHPs) typically do not provide direct treatment for individuals with diagnosed mental illnesses. Psychiatric care is predominantly managed at secondary and tertiary levels by specialists. PHPs mainly perform basic screening and referral. Consequently, most participants had limited or no direct clinical involvement with patients suspected of having or diagnosed with mental disorders.

### 2.2. Study Participants

Participants included primary healthcare professionals practicing in primary healthcare in Muscat during the study period. The eligibility criteria required participants to be licensed physicians with at least one year of experience in their current position. This approach aimed to better assess mental health stigma and related behaviors among established professionals.

### 2.3. Inclusion and Exclusion Criteria for Study Participants

The study included primary healthcare physicians who were active in Muscat during the study period. Eligible participants were licensed medical professionals with a minimum of one year of experience in their current primary care roles. Physicians who did not meet these criteria, including those with less than a year of experience, were excluded.

### 2.4. Data Collection Process

The research team contacted potential participants to arrange appointments at their respective workplaces. During these meetings, the purpose, confidentiality, and importance of anonymity were thoroughly explained to physicians. Following this, participants were asked to provide informed consent before proceeding. Each participant completed the study questionnaire in a private room, ensuring that the responses remained confidential. The paper questionnaire took approximately 20 min to complete, after which the research team collected the completed forms along with the signed consent documents.

### 2.5. Sample Size

Based on the total number of PCPs in Oman, and a 95% confidence interval and a 5% margin. The required sample size was calculated to be 191 at an 80% power level. The EpiInfo™ program was employed to determine the required sample size. The participants were chosen using a two-stage sampling procedure. First, Muscat was selected by chance as the cluster from nine possible regions in Oman. Additionally, within Muscat, a basic computer-generated list was compiled from a randomly selected sample of Working primary healthcare physicians.

### 2.6. Outcome Measures

The study used a structured questionnaire divided into two main sections: Demographic and Clinical Data and 12-Item Perceived Stigma. These are detailed below including the development of 12 items, Physicians Perceived Stigma Scale (PPSS).

### 2.7. Demographic and Clinical Data

The first section collected demographic information, including age, gender, years of practice, and specialty. Additionally, participants reported having episodes of depressive symptoms and the frequency of time off work due to mental health concerns. Help-seeking behaviors were assessed using questions regarding preferred sources of support during times of stress or stress, barriers to seeking help, and tendencies toward the self-prescription of medications.

### 2.8. The Physicians’ Perceived Stigma Scale (PPSS) 12-Item

The PPSS was developed for the present study, as the existing literature was deemed inadequate to capture the broader aspect of perceived stigma among physicians in Oman.

The scale was designed to evaluate four key domains: devaluation, the tendency to undervalue the abilities of physicians with mental health issues; discrimination, behavior, or intention to treat physicians differently due to their mental health status; endorsement of stereotypes, agreement with common stereotypes about mental illness; and secrecy, the perceived or expressed need for individuals to conceal their mental health struggles [3,10,11]. The initial phase involved defining the objectives and scope of the scale, followed by item generation based on a review of existing stigma measurement tools [12,13,14,15] and insights from interviews with physicians. A preliminary pool of 20 items was created, capturing context-specific relevant to the target population. As shown in Table 1, 12 items were deemed relevant for the present target population with the following constructs: (1) Devaluation: Items like #2, #3, #6, and #7 reflect the tendency to undervalue physicians with mental health issues, viewing them as less competent or trustworthy; (2) Discrimination: Items #5, #8, #10, and #12 suggest discrimination in employment and personal situations based on a physician’s mental health history; (3) Endorsement of Stereotypes: Items #4 and #9 show acceptance of generalized negative characteristics attributed to individuals with mental illness., and (4) Secrecy: Items like #1 and #11 hint at a perceived need to conceal mental health struggles to avoid negative treatment or judgment.

The responses were classified into three groups according to the participant’s agreement with the stigma statements: (if) Low Perceived Stigma Group: participants who agreed with fewer than half of the negative statements; (<6 questions) (ii) Intermediate Perceived Stigma Group: participants who agreed with six to eight negative statements. (6–8 questions); and (iii) High Perceived Stigma Group: participants who agreed with three-quarters or more of the negative statements. (9–12 questions). The 12 items that directly assessed stigma were used to form three categories of responses based on the number of negative statements with which respondents agreed and positive statements with which they disagreed. The categories grouped those who agreed with less than half of the statements (referred to as the ‘low perceived stigma’ group) and those who agreed with three-quarters or more of the statements (referred to as the ‘high perceived stigma group’). The ‘intermediate’ group agreed with six or more statements but fewer than nine. This ordinal classification has been used rather than a relative scaling method because it preserves a sense of the extent of professional training.

To ensure the validity of the content, a panel of five subject matter experts, including mental health professionals and scale development specialists, reviewed the elements for relevance, clarity, and comprehensiveness. Using a 4-point scale, experts rated each item’s relevance to the domain and items with a Content Validity Index (CVI) score of 0.80 or higher were retained. This process refined the scale to 12 items. A 6-point Likert scale format was chosen to force respondents to take a stance on each item, avoiding a neutral midpoint.

The scale underwent cognitive interviewing and pilot testing with a sample of 30 physicians from various specialties and experience levels. During the cognitive interviews, participants were asked to complete the preliminary version of the scale using a think-aloud approach, followed by targeted verbal probing to assess how they interpreted each item and chose their responses. This process helped identify any ambiguous wording, societal expectations, workplace culture, or conceptual misunderstandings. Based on the feedback, minor revisions were made to improve the clarity and contextual relevance of the items

A larger sample of 100 participants was then used to assess the psychometric properties. Reliability testing indicated strong internal consistency, with a Cronbach’s alpha of 0.85. The validity was also evaluated by comparing the PPSS with the Mental Illness: Clinicians’ Attitudes Scale—Version 4 (MICA-4), resulting in a strong positive correlation (r = 0.78, *p* < 0.01). The PPSS was correlated with the MICA-4, a 16-item validated instrument designed to measure clinicians’ attitudes toward mental illness. Each item is rated on a 6-point Likert scale: (Strongly Agree, Agree, Somewhat Agree, Disagree Strongly, Disagree, and Somewhat Disagree), with higher scores indicating more negative attitudes [12,16]. To assess test–retest reliability, a separate group of 50 physicians completed the PPSS-12 twice with a two-week interval, resulting in an intraclass correlation coefficient (ICC) of 0.88, indicating excellent reliability over time. These participants were excluded from the final study sample of 191 used for statistical analysis.

### 2.9. Statistical Analysis

Continuous variables were presented as mean, median, standard deviation and interquartile range, while categorical variables were presented as frequency and percentage. The association between two categorical variables was assessed using a Chi-square test (Fisher’s exact/likelihood ratio). A multivariate binary logistic regression analysis was performed to determine the independent effect of the association between perceived stigma and help-seeking behaviors (sources of help and barriers to help-seeking). Adjusted odds ratios (AORs) were presented with a 95% CI. A *p*-value less than 0.05 was considered statistically significant. All analysis was carried out in IBM SPSS Statistics (IBM Corp. Released 2022. IBM SPSS Statistics for Windows, version 29.0. Armonk, NY, USA: IBM Corp.).

### 2.10. Ethical Considerations

This study adhered to the ethical principles outlined in the Declaration of Helsinki, ensuring confidentiality, autonomy, responsible data management, and the acquisition of informed consent from participants. The ethical approval number (MH/DGPS/M6154) was obtained from the Ministry of Health, Oman, before the start of the study, approval date 12 January 2023. The study was reported using the STROBE reporting guidelines.

## 3. Results

### 3.1. Professional and Sociodemographic Characteristics

As shown in Table 2, the study included 191 participants, most of whom were female (78.5%) and between 30–39 years (49.7%). Most of the participants were Omani nationals (71.2%) and married (87.4%). A significant proportion had children (72.3%), while 41.4% had received official training as family physicians. Regarding professional roles, most were medical officers (77%), and only a small fraction served as consultants (2.6%).

### 3.2. Clinical Characteristics and Help-Seeking Behaviors of the Participants

The clinical characteristics of the participants are presented in Table 3. More than half of the participants (57.6%) reported experiencing an episode of depression in the last five years, yet only 21.8% sought professional help. Most of the participants (91.1%) reported taking fewer than seven sick leave days in the past year. When experiencing emotional or mental health problems, participants were more likely to turn to family or friends (37.2%) and less likely to seek the help of a mental health specialist at a different institute (3.7%). Furthermore, 24.6% reported that they would not seek help at all. The most common reason for not seeking help was a lack of belief in medication or psychotherapy (23%), followed by concerns about confidentiality (19.9%) and fear that others would view them less favorably if they sought treatment (18.3%).

### 3.3. Perceptions of Stigmatization

For the purpose of clearer interpretation and comparison of stigma endorsement across items, responses were later dichotomized into two categories: (1) Slightly/Disagree/Strongly disagree and (2) Slightly/Agree/Strongly agree for tabular reporting. This split allowed us to identify and summarize the proportion of participants expressing any level of endorsement for stigma-related beliefs, which was more meaningful for public health implications and intervention planning.

The perceptions of stigmatization are summarized in Table 3. Most of the respondents, 58.6%, disagreed with the statement that most doctors would accept a person with mental illness as a close friend. Furthermore, 58.1% believed that once hospitalized for mental illness, the opinions of a doctor would be taken less seriously, and 56.5% believed that such a doctor would face career promotion challenges. Furthermore, 78.5% of the respondents expressed reluctance to marry someone treated for mental or emotional illness. However, 62.3% disagreed with the statement that entering a psychiatric hospital is a sign of personal failure.

### 3.4. Association Between Stigma and Socio-Demographic Characteristics

As shown in Table 4, the association between perceived stigma and sociodemographic or professional characteristics was not statistically significant in most variables. For example, no significant differences were observed by age group, sex, marital status, or years of practice (*p* > 0.05).

### 3.5. Association Between Stigma Perception and Clinical Characteristics

The relationship between stigma and clinical characteristics is detailed in Table 5. A history of depressive episodes, the number of sick leave days, and previous self-prescription of medications did not show a significant association with perceived stigma (*p* > 0.05). 

### 3.6. Relationship Between Perception of Stigma and Help-Seeking Behaviors

Table 6 and Table 7 give the numbers and percentages of respondents who responded to the ‘help-seeking’ questions according to the category (low/intermediate or high) of responses to PPSS items.

Table 6 and Table 7 also give the adjusted odds ratios for different aspects of help-seeking, comparing the odds of responses of those in the ‘low/intermediate stigma’ group with those in the ‘high stigma’ group. These odds ratios were adjusted for age, sex, and history of depressive episodes.

Among potential sources of help, those in the high-perceived stigma group were significantly more likely to turn to family or friends (adjusted OR = 2.873; 95% CI = 1.345–6.138; *p* = 0.006). Other sources of help and barriers to seeking care did not produce statistically significant associations.

#### Help-Seeking Barriers

As illustrated in Table 7, the most cited barrier to help-seeking was a lack of belief in the effectiveness of medication or psychotherapy, reported by 23.1% of the participants. Other barriers, including concerns about confidentiality (14.3%) and fear of documentation in academic records (8.6%), did not differ significantly between stigma groups (*p* > 0.05).

## 4. Discussion

This study illuminates the sociodemographic, professional, and clinical characteristics of healthcare providers, focusing on their own perception of their perceptions of mental health, their mental health stigma, and help-seeking behaviors. The findings highlight significant challenges and offer a platform to reflect on global international trends and the Omani cultural factors that influence these challenges.

The high prevalence of self-reported depressive episodes (57.6%) among study participants aligns with global trends that indicate healthcare workers are at increased risk for mental health disorders. Studies in Oman and the United States have similarly reported elevated rates of burnout, depression, and anxiety among physicians and other healthcare professionals, often exacerbated by demanding work environments, long hours, and emotional stress from patient care responsibilities [17,18,19]. However, the proportion of participants who sought professional help (21.8%) is low, even compared to global averages. A review article has estimated help-seeking rates among physicians with mental disorders, with 36% of healthcare providers with mental health issues seeking professional help, highlighting regional and cultural disparities in help-seeking behavior [20].

The reluctance to seek help observed in this study reflects cultural influences that are particularly pronounced in Middle Eastern and South Asian contexts, where the stigma surrounding mental illness remains deeply ingrained, and in Oman, psychiatrists also noticed the stigma towards mental health care institutes in the Omani community [21]. In our study, healthcare providers reported concerns about confidentiality (19.9%), fear of judgment (18.3%), and skepticism toward the effectiveness of psychotherapy or medication (23%), which are consistent with findings from studies in Saudi Arabia and India, where healthcare providers reported similar barriers [22,23]. In contrast, Western countries, known for their progressive mental health policies and robust support systems, report higher rates of professional help-seeking among healthcare workers, likely due to reduced stigma and greater access to mental health resources [24].

Cultural expectations also play a significant role in shaping attitudes toward mental health. In many collectivist societies [25], including Oman, individuals often rely on family or close social networks for support, as seen in this study, where participants turned predominantly to family and friends (37.2%) rather than professional mental health services. This contrasts with findings from individualistic cultures, such as those in North America and western Europe, where there is greater emphasis on personal agency and professional intervention for mental health issues [24].

The predominance of female participants (78.5%) in this study mirrors global trends, indicating an increasing representation of women in healthcare professions. However, women in healthcare are also disproportionately affected by work–life balance challenges, which contribute to higher rates of burnout and mental health issues. A study from Saudi Arabia has highlighted that women physicians are less likely to seek help for mental health concerns due to fear of stigma, neglect by the healthcare administration, and potential career repercussions [26], similar to the concerns expressed by the participants in this study.

The professional hierarchy within the healthcare system also plays a critical role. The limited representation of senior professionals, such as consultants (2.6%), and the predominance of medical officers (77%) suggest potential disparities in professional development opportunities, which can contribute to job dissatisfaction and mental health challenges [27].

The findings emphasize the need for targeted interventions to address stigma and promote awareness of mental health in healthcare settings. Lessons can be drawn from international initiatives, such as the “Physician Health Program” in Canada, which provides customized confidential support and counseling services customized for healthcare providers [28]. Adopting similar models in Oman while considering cultural sensitivity could help bridge the gap in mental health care for healthcare workers.

Efforts to reduce stigma must also involve broader societal changes. Public education campaigns, similar to Australia’s “R U OK?” initiative, could normalize discussions about mental health and encourage help-seeking behaviors [29]. Additionally, workplace interventions, such as mindfulness training, peer support groups, and resilience-building workshops, have shown promise in improving mental health outcomes among healthcare providers in high-stress environments [30].

The findings demonstrate mixed stigmatizing attitudes toward mental illness among medical professionals, including skepticism about the credibility of colleagues with a history of mental health challenges. This aligns with international studies that report similar professional stigma. For example, research from Western countries indicates that stigma persists in healthcare professions despite efforts to promote mental health awareness, and professionals often perceive colleagues with mental illness as less competent or trustworthy [31]. However, in some regions, stigma is even more pronounced due to cultural norms that emphasize personal resilience and avoidance of vulnerability, as observed in certain Middle Eastern and Asian contexts [25].

Interestingly, the openness shown by respondents in treating colleagues with mental illness equitably aligns with trends in high-income countries where stigma reduction campaigns and anti-stigma programming have been moderately successful [32]. However, the reluctance to accept people with mental health challenges in personal roles, such as marriage or childcare, reflects deeper cultural and professional stigmas that are also evident in the Omani culture, as shown by the findings of this study.

The absence of significant associations between stigma and demographic factors in this study mirrors findings in international literature, where stigma is often reported to transcend age, gender, or professional status. However, studies from countries with stronger mental health advocacy frameworks, such as the West, report slight variations, with younger and more recently trained professionals showing reduced stigma levels compared to their older counterparts. This difference may highlight the impact of education and generational changes in attitudes toward mental health [33].

In addition to demographic factors, the study explored how stigma levels influenced perceived barriers to help-seeking and preferred sources of support. The analysis revealed no statistically significant associations between stigma levels and common barriers to help-seeking, including concerns about confidentiality, time constraints, or fear of negative career implications. Although trends suggested that those with higher stigma may be slightly more concerned about confidentiality or fear of judgment [34], these associations did not reach significance. This finding suggests that stigma may not act independently but rather interact with broader institutional and cultural factors influencing help-seeking behavior [34].

Notably, the analysis of sources of help revealed that participants with higher stigma were significantly more likely to rely on informal support systems, such as family or friends (OR 2.87, *p* = 0.006), rather than formal mental health professionals. This reinforces the influence of cultural norms within collectivist societies, such as Oman, where emotional support is often sought within trusted personal networks. It also points to potential mistrust or discomfort with formal mental health systems among those harboring more stigmatizing attitudes [35].

Lastly, no significant associations were observed between stigma and participants’ previous experiences with depression, sick leave, or self-prescribing psychiatric medications. Similarly, the willingness to seek professional help for emotional or mental health problems was not significantly associated with stigma levels. These findings suggest that stigma, while influential in shaping general attitudes, may not always translate into observable differences in behavior, particularly in contexts where other factors such as accessibility, institutional support, or personal coping strategies play a mediating role [36].

### Strengths, Limitations, and Future Directions

This study’s strengths lie in its detailed exploration of sociodemographic and clinical characteristics, which provides a detailed understanding of the challenges facing healthcare providers. However, the reliance on self-reported data introduces potential biases such as social desirability bias, and the cross-sectional design limits the ability to establish causality. Future research should consider longitudinal designs and cross-cultural comparisons to identify effective strategies for addressing mental health challenges and reducing stigma among healthcare providers. The sample size (n = 191) satisfies some exploratory factor analysis (EFA) criteria, including 5–10 participants per item; nevertheless, other psychometric experts advocate for a minimum of 300 participants to guarantee the stability and interpretability of the component structure [37]. Consequently, this study concentrated on establishing initial reliability and external validity for the scale, and we advocate for future research with larger and more diverse populations to do comprehensive psychometric validation, including exploratory factor analysis (EFA) and confirmatory factor analysis (CFA) [37]. To further verify the scale and confirm its factorial structure, these will be essential in future investigations with bigger and more varied populations. The generalizability and robustness of psychometric assessment may be further improved by increasing the diversity of the sample in terms of both geography and specialty. The high proportion of women in the sample (approximately three-quarters) reflects the actual gender distribution within the primary healthcare physician workforce in Oman. The healthcare sector in Oman has undergone a process of feminization over the past decade, with a growing number of women entering and dominating roles in healthcare. This trend is attributed to cultural and social factors, including reduced Omani males’ interest in healthcare professions. Therefore, the sample composition is representative of the actual PHP population.

## 5. Conclusions

This study highlights significant mental health challenges and the widespread influence of stigma on help-seeking behaviors among primary healthcare providers. When compared with international trends and reflecting on cultural differences, it becomes evident that addressing these challenges requires a multifaceted and culturally sensitive approach. Initiatives to reduce stigma, improve access to mental health resources, and promote supportive workplace environments are critical. Such efforts will not only improve the well-being of healthcare providers but also ensure the sustainability of healthcare systems worldwide.

## Figures and Tables

**Table 1 ijerph-22-00983-t001:** Perceptions of stigmatization were tapped by the newly developed Physicians’ Perceived Stigma Scale—12-item questionnaires.

Item	Slightly Disagree/Disagree/Strongly Disagree	Slightly Agree/Agree/Strongly Agree
	n (%)	n (%)
1. Most doctors would accept a person who has a mental illness as a close friend	112 (58.6)	79 (41.4)
2. Most doctors believe that a doctor who has been hospitalised for mental illness is just as intelligent as other doctors	86 (45.0)	105 (55.0)
3. Most physicians believe that a doctor with a mental illness is just as trustworthy as other doctors	87 (45.5)	104 (54.5)
4. Most doctors would accept a person who has fully recovered from mental illness as a teacher of young children in a public school	83 (43.5)	108 (56.5)
5. Most employers will reject the application for promotion of a doctor who has had a mental / emotional problem in favor of another applicant	83 (43.5)	108 (56.5)
6. Most doctors think less of another doctor after being hospitalized for a mental illness	89 (46.6)	102 (53.4)
7. Once they know that a doctor was in a mental hospital for mental / emotional problems, most doctors will take his or her opinions less seriously	80 (41.9)	111 (58.1)
8. Most employers will hire a doctor who has been hospitalised for mental illness if he or she is qualified for the job	77 (40.3)	114 (59.7)
9. Most physicians believe that entering a psychiatric hospital is a sign of personal failure	119 (62.3)	72 (37.7)
10. Most doctors will not hire someone who has been hospitalized for a serious mental illness to take care of their child, even if he or she had been well for some time	51 (26.7)	140 (73.3)
11. Most physicians would treat a doctor who has been hospitalized for mental illness as they would treat anyone	56 (29.3)	135 (70.7)
12. Most doctors would be reluctant to marry someone who has been treated for mental/ emotional illness	41 (21.5)	150 (78.5)

**Table 2 ijerph-22-00983-t002:** Professional and sociodemographic characteristics of the participants.

	Variable	n (%)
Sex	Male	41 (21.5)
	Female	150 (78.5)
Age	18 to 29 years	36 (18.8)
	30 to 39 years	95 (49.7)
	40 to 49 years	48 (25.1)
	50 years and above	12 (6.4)
Nationality	Omani	136 (71.2)
	Non-Omani	55 (28.8)
Marital status	Married	167 (87.4)
	Single	19 (9.9)
	Divorced/Widowed	5 (2.7)
Do you have children?	Yes	138 (72.3)
	No	53 (27.7)
Did you receive formal training as a family physician?	Yes	79 (41.4)
	No	112 (58.6)
How long have you been practicing medicine	Less than 2 years	12 (6.3)
	2 to 5 years	59 (30.9)
	6 to 10 years	46 (24.1)
	11 to 15 years	28 (14.7)
	16 to 20 years	30 (15.7)
	more than 20 years	16 (8.4)
What is your job title?	Medical officer	147 (77.0)
	Specialist	39 (20.4)
	Consultant	5 (2.6)

**Table 3 ijerph-22-00983-t003:** Clinical Characteristics and Help-Seeking Behaviors of the Participants.

Variable	n (%)
Have you had an episode of depression in the past 5 years	
Yes	110 (57.6)
No	81 (42.4)
If yes, did you seek help (n = 110)	
Yes	24 (21.8)
No	86 (78.2)
How many sick leave days did you take last year?	
0 to 7 days	174 (91.1)
8 to 14 days	14 (7.3)
more than 14 days	3 (1.6)
If you have emotional or mental health problems, to what sources of help would you turn to?	
Family or Friend	71 (37.2)
work colleagues	16 (8.4)
Mental health specialist working in the same institute where you are working	22 (11.5)
Mental health specialist working in a different institute than where you are working	7 (3.7)
Private psychiatry clinic	28 (14.7)
would not seek help at all	47 (24.6)
If you have emotional or mental health problems, would any of the following be a reason not to seek help?	
Concerned about confidentiality	38 (19.9)
lack of time	23 (12.0)
Fear that it may hurt your career progression	23 (12.0)
Fear that others will look at you less favorably if they know that you are receiving treatment for a mental problem	35 (18.3)
Fear of documentation on your academic record	11 (5.8)
Fear of an unwanted intervention	17 (8.9)
I do not believe in medication or psychotherapy	44 (23.0)
If you have a serious mental or emotional problem, would you seek help	
I would go	92 (48.2)
I would probably go	64 (33.5)
I would probably not go	25 (13.1)
I would not go	10 (5.2)
Have you ever self-prescribed any of these medication anti-depressants	
Yes	18 (9.4)
No	173 (90.6)
Have you ever self-prescribed any of these medication anxiolytics	
Yes	11 (5.8)
No	180 (94.2)
Have you ever self-prescribed any of these medications sleeping medication	
Yes	25 (13.1)
No	166 (86.9)

**Table 4 ijerph-22-00983-t004:** The Association between stigma perception and socio-demographic-professional characteristics.

Variable	Variation of Stigma		*p*-Value
	Low/Intermediate	High	
	n (%)	n (%)	
Age groups			
18–29 years	29 (80.6)	7 (19.4)	0.995
30–39 years	78 (82.1)	17 (17.9)	
40–49 years	39 (81.3)	9 (18.8)	
≥50 years	10 (83.3)	2 (16.7)	
Sex			
Male	32 (78.0)	9 (22.0)	0.500
Female	124 (82.7)	26 (17.3)	
Nationality			
Omani	115 (84.6)	21 (15.4)	0.147
Non-Omani	41 (74.5)	14 (25.5)	
Marital status			
Married	136 (81.4)	31 (18.6)	0.951
Single	16 (84.2)	3 (15.8)	
Divorced/Widowed	4 (80.0)	1 (20.0)	
Do you have children?			
Yes	114 (82.6)	24 (17.4)	0.677
No	42 (79.2)	11 (20.8)	
Did you receive formal training as a family physician?			
Yes	61 (77.2)	18 (22.8)	0.190
No	95 (84.8)	17 (15.2)	
How long have you been practicing medicine			
<2 years	11 (91.7)	1 (8.3)	0.680
2–5 years	47 (79.7)	12 (20.3)	
6–10 years	38 (82.6)	8 (17.4)	
11–15 years	24 (85.7)	4 (14.3)	
16–20 years	22 (73.3)	8 (26.7)	
>20 years	14 (87.5)	2 (12.5)	
Job title			
Medical officer	116 (78.9)	31 (21.1)	0.114
Specialist	36 (92.3)	3 (7.7)	
Consultant	4 (80.0)	1 (20.0)	
*Test: Chi-square test (Fisher’s exact/Likelihood ratio)*			

**Table 5 ijerph-22-00983-t005:** The association between sociodemographic and professional characteristics and levels of stigma toward mental illness.

Variable	Stigma		*p*-Value
	Low/Intermediate	High	
	n (%)	n (%)	
Did you have an episode of depression in the past 5 years			
Yes	89 (80.9)	21 (19.1)	0.851
No	67 (82.7)	14 (17.3)	
How many sick days did you take last year			
0 to 7 days	143 (82.2)	31 (17.8)	0.778
8 to 14 days	11 (78.6)	3 (21.4)	
more than 14 days	2 (66.7)	1 (33.3)	
If you have emotional or mental health problems, to what sources of help would you turn to?			
Family or Friend	51 (71.8)	20 (28.2)	0.090
work colleagues	14 (87.5)	2 (12.5)	
Mental health specialist working in the same institute where you are working	20 (90.9)	2 (9.1)	
Mental health specialist working in a different institute than where you are working	7 (100)	0 (0)	
Private psychiatry clinic	24 (85.7)	4 (14.3)	
would not seek help at all	40 (85.1)	7 (14.9)	
If you have emotional or mental health problems, would any of the following be a reason not to seek help?			
Concerned about confidentiality	33 (86.8)	5 (13.2)	0.884
lack of time	18 (78.3)	5 (21.7)	
Fear that it may have a negative effect on your career progression	20 (87.0)	3 (13.0)	
Fear that others will look at you in a less favorable way if they know that you receive treatment for mental problem	27 (77.1)	8 (22.9)	
Fear of documentation on your academic record	8 (72.7)	3 (27.3)	
Fear of an unwanted intervention	14 (82.4)	3 (17.6)	
I do not believe in medication or psychotherapy	36 (81.8)	8 (18.2)	
If you have serious mental or emotional problems, would you seek help?			
Yes	128 (82.1)	28 (17.9)	0.810
No	28 (80.0)	7 (20.0)	
Have you ever self-prescribed any of these medication anti-depressants			
Yes	14 (77.8)	4 (22.2)	0.748
No	142 (82.1)	31 (17.9)	
Have you ever self-prescribed any of these medication anxiolytics			
Yes	10 (90.9)	1 (9.1)	0.692
No	146 (81.1)	34 (18.9)	
Have you ever self-prescribed any of these medications sleeping medication			
Yes	21 (84.0)	4 (16.0)	1.000
No	135 (81.3)	31 (18.7)	
*Test: Chi-square test (Fisher’s exact/Likelihood ratio)*			

**Table 6 ijerph-22-00983-t006:** The association between sources of help-seeking and levels of stigma toward mental illness.

Sources of Help	Stigma		*p*-Value
	Low/Intermediate	High	
	n (%)	n (%)	
Family or Friend			
Yes, n (%)	51 (32.7)	20 (57.1)	
Adjusted OR (95% CI)	1 (Reference)	2.873 (1.345–6.138)	0.006 *
work colleagues			
Yes, n (%)	14 (9.0)	2 (5.7)	
Adjusted OR (95% CI)	1 (Reference)	0.613 (0.132–2.851)	0.532
Mental health specialist working in the same institute where you are working			
Yes, n (%)	20 (12.8)	2 (5.7)	
Adjusted OR (95% CI)	1 (Reference)	0.429 (0.095–1.947)	0.273
Mental health specialist working in a different institute than where you are working			
Yes, n (%)	7 (4.5)	0 (0)	
Adjusted OR (95% CI)	1 (Reference)	*NE*	*NE*
Private psychiatry clinic			
Yes, n (%)	24 (15.4)	4 (11.4)	
Adjusted OR (95% CI)	1 (Reference)	0.681 (0.216–2.149)	0.512
Would not seek help at all			
Yes, n (%)	40 (25.6)	7 (20.0)	
Adjusted OR (95% CI)	1 (Reference)	0.703 (0.283–1.745)	0.447

* Statistically significant, NE—Not Estimated, ORs adjusted for age, gender, and History of depressive episode.

**Table 7 ijerph-22-00983-t007:** The association between perceived barriers to help-seeking and levels of stigma toward mental illness.

Barriers to Help-Seeking	Stigma		*p*-Value
	Low/Intermediate	High	
	n (%)	n (%)	
Concerned about confidentiality			
Yes, n (%)	33 (21.2)	5 (14.3)	
Adjusted OR (95% CI)	1 (Reference)	0.625 (0.222–1.756)	0.373
Lack of time			
Yes, n (%)	18 (11.5)	5 (14.3)	
Adjusted OR (95% CI)	1 (Reference)	1.356 (0.458–4.011)	0.582
Fear that it may hurt your Career progression			
Yes, n (%)	20 (12.8)	3 (8.6)	
Adjusted OR (95% CI)	1 (Reference)	0.613 (0.170–2.209)	0.454
Fear that others will look at you less favorably if they know that you are receiving treatment for a mental problem			
Yes, n (%)	27 (17.3)	8 (22.9)	
Adjusted OR (95% CI)	1 (Reference)	1.408 (0.572–3.464)	0.456
Fear of documentation on your academic record			
Yes, n (%)	8 (5.1)	3 (8.6)	
Adjusted OR (95% CI)	1 (Reference)	1.745 (0.435–7.006)	0.432
Fear of an unwanted intervention			
Yes, n (%)	14 (9.0)	3 (8.6)	
Adjusted OR (95% CI)	1 (Reference)	0.995 (0.266–3.725)	0.994
I do not believe in medication or psychotherapy			
Yes, n (%)	36 (23.1)	8 (22.9)	
Adjusted OR (95% CI)	1 (Reference)	0.963 (0.398–2.327)	0.933

ORs adjusted for age, gender, and history of depressive episodes.

## Data Availability

The raw data supporting the conclusions of this article will be made available by the authors upon request. Because of the sensitive nature of the collected data related to mental health stigma, and to ensure full compliance with institutional and privacy regulations, moreover, given that our sample comprises a small, well-defined group of physicians working in primary care in Muscat City, there is a possible risk of reidentification. Nonetheless, the raw data supporting the conclusions of this article will be made available by the authors on request.

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
