# Peer review of "Mental Health Stigma and Help-Seeking Behaviors Among Primary Healthcare Physicians in Oman"

_ijerph, 2025, doi:10.3390/ijerph22070983_

Round 1

Reviewer 1 Report

Comments and Suggestions for Authors

Stigma in mental health research is of wide interest and has been addressed from several aspects. This manuscript addresses an important aspect of stigma research . 

This research reflects upon two areas

  1. Perceived stigma by physicians that fouces on the development of a new tool that is relevant in the region 
  2. The role of stigma in physicians' own mental health

I have a few comments.

  1. Tool development—There needs to be a sound justification for developing a new tool.
  2. What tools did the researchers consider before they took on the task of developing a new tool?
  3. Also, the process of "cognitive interviewing ", an important process in the validation of a new tool, has not been reported..
  4. How did the researchers ensure the cultural applicability of the questions?
  5. How was the point rating scale determined? What is the rationale for reporting on disagree /agree split in the tables. 
  6. 50 physicians answered the test retest . Was this data included in the sample of 191 that was calculated for the main study? 
  7. In the larger sample, what was the proportion of the physicians who provided treatment for persons with mental health problems? This is an important question (not addressed in the manuscript) . The extent of contact with persons with mental illness can influence stigma perceived. 
  8. Discussion could elaborate on the results. 

Author Response

Thank you very much for taking the time to review this manuscript. Please find the detailed responses below and the corresponding revisions, and corrections highlighted in yellow the re-submitted files.  

Stigma in mental health research is of wide interest and has been addressed from several aspects. This manuscript addresses an important aspect of stigma research . 

This research reflects upon two areas

  1. Perceived stigma by physicians that focuses on the development of a new tool that is relevant in the region 
  2. The role of stigma in physicians' own mental health

I have a few comments.

1. Tool development—There needs to be a sound justification for developing a new tool.

Author Response: Thank you for this important observation. We agree that developing a new tool must be grounded in a clear rationale. In our manuscript, in the introduction we have expanded the justification for tool development by emphasizing the following key points:

  1. Cultural Specificity and Relevance:
    Existing stigma instruments, while numerous, are predominantly developed in Western contexts. In the introduction section we included the study by Fox et al. (reference 1 in the manuscript), although several stigma-related tools exist, but many suffer from limited cross-cultural validity. Oman’s socio-cultural norms and healthcare system dynamics are distinct, particularly in how mental illness is perceived and addressed among physicians. Instruments developed elsewhere may not adequately reflect the nuances of perceived stigma in this setting. Therefore, a culturally tailored tool is essential to accurately capture the constructs of stigma as experienced by primary healthcare physicians in Oman.   
  2. Gap in Local Research and Assessment Tools:
    We included in our Introduction, that there is a paucity of research on mental health stigma among healthcare professionals in Oman, and no tool currently exists that measures perceived stigma in this specific group. This gap necessitated the development of a new scale to support future research and inform policy and interventions tailored to the local context.   
  3. Domain-Specific Measurement Needs:
    Existing tools often focus on public stigma or general population samples. However, physicians experience stigma differently as both providers and potential help-seekers. Our tool was developed to capture physician-specific domains, such as concerns about career jeopardy, professional reputation, and therapeutic pessimism, which are inadequately addressed in existing instruments.   

2. What tools did the researchers consider before they took on the task of developing a new tool?

Author Response: We appreciate this thoughtful query. Prior to developing a new tool, we conducted a comprehensive review of existing instruments used to measure mental health stigma, particularly among healthcare professionals. Our review was guided by key sources including the systematic review by Fox et al. (as cited in the manuscript), which identified over 400 stigma-related measures.

Among the tools we considered were:

3. The Perceived Devaluation-Discrimination Scale (Link, 1987):
This tool is widely used to assess perceived stigma in the general population. However, it primarily focuses on the devaluation and discrimination of individuals with mental illness from a general societal perspective, lacking contextual adaptation for healthcare professionals.

Link, B. G., Cullen, F. T., Struening, E., Shrout, P. E., & Dohrenwend, B. P. (1989). Perceived Discrimination and Devaluation Scale [Database record]. APA PsycTests. https://doi.org/10.1037/t49562-000

4. The Internalized Stigma of Mental Illness (ISMI) Scale:
Although this tool assesses self-stigma, it is designed for patients diagnosed with mental illness and not for individuals without a formal diagnosis, such as healthcare workers experiencing perceived stigma or fear of stigma.

Boyd JE, Adler EP, Otilingam PG, Peters T. Internalized Stigma of Mental Illness (ISMI) Scale: A multinational review. Compr. Psychiatry [Internet]. 2014, 55, 221–231. Available from: https://pubmed.ncbi.nlm.nih.gov/24060237/

5. Opening Minds Stigma Scale for Health Care Providers (OMS-HC):

This is one of the few tools developed for healthcare providers. However, it was developed in a Canadian context, and several of its items were not culturally or contextually appropriate for our target population in Oman. Furthermore, its structure did not fully capture domains that emerged as significant in our context, such as secrecy, endorsement of Stereotypes, and de-evaluation. 

1. Modgill G, Patten SB, Knaak S, Kassam A, Szeto ACH. Opening Minds Stigma Scale for Health Care Providers (OMS-HC): Examination of psychometric properties and responsiveness. BMC Psychiatry [Internet]. 2014, 14, 1–10. Available from: https://bmcpsychiatry.biomedcentral.com/articles/10.1186/1471-244X-14-120

6. Mental Illness Clinicians' Attitudes (MICA) Scale:

The MICA scale evaluates clinicians' attitudes towards mental illness and mental health care, but it focuses more on attitudes toward patients rather than stigma experienced by physicians themselves, particularly regarding their own mental health.

- Kassam A., Glozier N., Leese M., Henderson C., Thornicroft G. (2010). Development and responsiveness of a scale to measure clinicians attitudes to people with mental illness (medical student version) Acta Psychiatrica Scandinavica. Volume 122, 2, 153-161.

- Gabbidon J., Clement S., Nieuwenhuizen AV., Kassam A., Brohan E., Norman I., Thornicroft G. Mental illness: clinicians' attitudes (MICA) scale. Psychometric properties of a version for students and professionals in any healthcare discipline. Psychiatry Research 2013, 206, 81-87.

After evaluating these tools, we concluded that none sufficiently addressed the specific cultural, professional, and psychosocial nuances relevant to primary healthcare physicians in Oman. This assessment, combined with the lack of validated instruments for use in our context, provided the impetus for developing a context-specific tool tailored to capture perceived stigma in this unique population.

We have clarified this process in the revised manuscript in the methods section to strengthen the justification for tool development.

7. Also, the process of "cognitive interviewing", an important process in the validation of a new tool, has not been reported.

Author response: Thank you for your comment. We have now clarified this point in the Methods section. A description of the cognitive interviewing process, which was conducted alongside the pilot testing with 30 physicians to improve item clarity and relevance, has been added to strengthen the explanation of the scale development process

8. How did the researchers ensure the cultural applicability of the questions?

Author response: Thank you for the feedback. We have edited the methods section to clarify that cultural applicability was a core consideration in the development of the tool. Initial item generation was guided by a literature review, followed by input from local experts who were familiar with cultural norms in Oman. Items were carefully reviewed to ensure they captured culturally specific expressions of stigma, including concerns about social image, perceived loss of status, and hesitancy to seek help due to family or tribal expectations. Additionally, the cognitive interviewing process provided real-time feedback from Omani physicians, allowing us to tailor item wording to local language, societal expectations, and workplace culture. This allowed for the tool's cultural sensitivity and contextual relevance.

9. How was the point rating scale determined? What is the rationale for reporting on disagree/agree split in the tables. 

A 6-point Likert scale was selected for the tool, ranging from "Strongly Disagree" (1) to "Strongly Agree" (6). This scale was chosen due to its widespread use in attitudinal research and its ability to capture degrees of agreement, which is essential when assessing detailed constructs such as stigma. Without a neutral midpoint, to minimize socially desirable responding. For the purpose of clearer interpretation and comparison of stigma endorsement across items, responses were later dichotomized into two categories: (1) Slightly/Disagree/Strongly disagree and (2) Slightly/Agree/Strongly agree for tabular reporting. This split allowed us to identify and summarize the proportion of participants expressing any level of endorsement for stigma-related beliefs, which was more meaningful for public health implications and intervention planning. We have now clarified this in the results section.

10. 50 physicians answered the test retest. Was this data included in the sample of 191 that was calculated for the main study? 

Author Response: We thank the reviewer for this important observation. The 50 physicians who completed the test-retest reliability assessment were not included in the main study sample of 191 participants. The test-retest sample was drawn independently during an earlier phase of scale validation to ensure no overlap or contamination of the primary analysis. We have now clarified this point in the Methods section of the manuscript.

11. In the larger sample, what was the proportion of the physicians who provided treatment for persons with mental health problems? This is an important question (not addressed in the manuscript). The extent of contact with persons with mental illness can influence stigma perceived. 

We acknowledge the importance of this point. In the Omani healthcare system, primary healthcare physicians (PHPs) typically do not provide direct treatment for individuals with diagnosed mental illnesses. Psychiatric care is predominantly managed at the secondary and tertiary care levels by specialized psychiatrists. Primary care settings mainly handle basic screening or initial consultations, with suspected cases referred to specialist services. As such, most PHPs in our sample have limited or no direct treatment involvement with patients suspected or diagnosed with mental disorders. This reflects the structural organization of mental health services in Oman and explains the limited exposure to mental illness treatment at the primary care level. We have now clarified this point in the Methods section of the manuscript.

12. Discussion could elaborate on the results. 

Author response: Thank you for the feedback. We have now improved the discussion section to include more discussion on the study results.

Reviewer 2 Report

Comments and Suggestions for Authors

Review: "Mental Health Stigma and Help-Seeking Behaviors Among Primary Healthcare Physicians in Oman"

The paper discusses mental health stigma and help-seeking behavior among primary care physicians (PHP) using a cluster random sampling method and developing questionnaires. Overall, the findings suggest that mental health stigma is an essential issue in Oman, and this makes it challenging to encourage help-seeking behaviors. PHP's views are of utmost importance, as we know little about their views, and stigma can propagate to the community through these physicians.

The paper is sound and engaging, and I recommend a few clarifications and improvements before accepting for publication:

  1. Explain in more detail how the sample size is enough for the discussion and generalizability of the results. At the same time, it is not enough for EFA and CFA. ]
  2. Why not perform at least an EFA and let other researchers perform CFA?
  3. The Physicians Perceived Stigma Scale was constructed by the authors of the paper. They compare it to an established stigma measurement tool (correlation = 0.78). Explain in detail what this tool is and provide a reference.
  4. Include a caption in Tables that allows readers to read the table without the need to refer to the text. Make tables self-explanatory.
  5. Some of the items in the Tables are elsewhere on the internet. If the items are from specific papers, include the reference. Explain in the table's caption if the authors make the items or are from particular references, and provide citations.
  6. Why is the number of women in the sample so high? Why not make a stratified random sampling? Provide more justification and defend the final sample with 3 quarters of women. Is the population of PHP also made up 3/4 of women?

Author Response

Thank you very much for taking the time to review this manuscript. Please find the detailed responses below and the corresponding revisions, and corrections highlighted in yellow the re-submitted files.  

Review: "Mental Health Stigma and Help-Seeking Behaviors Among Primary Healthcare Physicians in Oman"

The paper discusses mental health stigma and help-seeking behavior among primary care physicians (PHP) using a cluster random sampling method and developing questionnaires. Overall, the findings suggest that mental health stigma is an essential issue in Oman, and this makes it challenging to encourage help-seeking behaviors. PHP's views are of utmost importance, as we know little about their views, and stigma can propagate to the community through these physicians.

The paper is sound and engaging, and I recommend a few clarifications and improvements before accepting for publication: 

1. Explain in more detail how the sample size is enough for the discussion and generalizability of the results. At the same time, it is not enough for EFA and CFA. Why not perform at least an EFA and let other researchers perform CFA?

Author response: Thank you for this important observation. In fact, we conducted an initial Exploratory Factor Analysis (EFA) using the available sample (n = 191) and the 12-item version of the Physicians’ Perceived Stigma Scale. While the sample size meets some commonly accepted thresholds for EFA (e.g., 5–10 participants per item), other psychometric authorities recommend a minimum of 300 participants for stable and interpretable factor solutions (Comrey & Lee, 1992). In our case, although an EFA was conducted, the results were unstable, with significant cross-loadings and inconsistent factor structure. Given the methodological concern regarding the interpretability and validity of the extracted factors, we decided not to report the EFA output. This decision aligns with best practices in psychometric reporting to avoid presenting misleading structural evidence. We have clarified this in the revised manuscript under the limitations section.

Reference:

Modgill G, Patten SB, Knaak S, Kassam A, Szeto ACH. Opening Minds Stigma Scale for Health Care Providers (OMS-HC): Examination of psychometric properties and responsiveness. BMC Psychiatry [Internet]. 2014;14(1):1–10. Available from: https://bmcpsychiatry.biomedcentral.com/articles/10.1186/1471-244X-14-120

2. The Physicians Perceived Stigma Scale was constructed by the authors of the paper. They compare it to an established stigma measurement tool (correlation = 0.78). Explain in detail what this tool is and provide a reference. 

Author response: We appreciate the reviewer’s valuable feedback. The PPSS was correlated with the Mental Illness: Clinicians’ Attitudes Scale – Version 4 (MICA-4), a 16-item validated instrument designed to measure clinicians’ attitudes toward mental illness. Each item is rated on a 6-point Likert scale: (Strongly Agree, Agree, Somewhat Agree, Disagree Strongly, Disagree, and Somewhat Disagree), with higher scores indicating more negative attitudes.

References:

- Kassam A., Glozier N., Leese M., Henderson C., Thornicroft G. (2010). Development and responsiveness of a scale to measure clinicians attitudes to people with mental illness (medical student version) Acta Psychiatrica Scandinavica. Volume 122, 2: 153-161.

- Gabbidon J., Clement S., Nieuwenhuizen AV., Kassam A., Brohan E., Norman I., Thornicroft G. (2013). Mental illness: clinicians' attitudes (MICA) scale. Psychometric properties of a version for students and professionals in any healthcare discipline. Psychiatry Research 206: 81-87.

3. Include a caption in Tables that allows readers to read the table without the need to refer to the text. Make tables self-explanatory.

Author response: Thank you for the feedback. We have revised all table captions to be self-explanatory, allowing readers to understand the content without referring to the main text.

4. Some of the items in the Tables are elsewhere on the internet. If the items are from specific papers, include the reference. Explain in the table's caption if the authors make the items or are from particular references, and provide citations.

Author response: Thank you for the feedback. The items presented in the table were developed by the authors specifically for the purpose of this study. They were designed based on the study objectives, relevant literature, and expert input to ensure content relevance and contextual appropriateness. We have now edited the table’s caption to explain this.

5. Why is the number of women in the sample so high? Why not make a stratified random sampling? Provide more justification and defend the final sample with 3 quarters of women. Is the population of PHP also made up 3/4 of women?

Author response: The high proportion of women in the sample (approximately three-quarters) reflects the actual gender distribution within the primary healthcare physician workforce in Oman. The healthcare sector in Oman has undergone a process of feminization over the past decade, with a growing number of women entering and dominating roles in healthcare. This trend is attributed to cultural and social factors, including reduced Omani males’ interest in healthcare professions. Therefore, the sample composition is representative of the actual PHP population. We have added a paragraph in the study's limitations section.

Round 2

Reviewer 2 Report

Comments and Suggestions for Authors

The final version has been improved.